# Synthesis of high-entropy hydride from the cantor alloy (fcc–CoCrFeNiMn) at extreme conditions

Konstantin Glazyrin [1] ✉, Kristina Spektor [1] ✉, Maxim Bykov [2], Paulo H. B. Carvalho[3], Weiwei Dong[1,14], Fritz Körmann[4,5,6], Asami Sano-Furukawa [7], Takanori Hattori [7], Doreen C. Beyer [8], Martin Sahlberg[3], Yuji Ikeda [4], Ji Hun Yu[9], Yang Sangsun[9], Jai-Sung Lee [10], Shrikant Bhat [1], Michael Hanfland[11], Blazej Grabowski [4], Sergiy Divinski [12] & Kirill V. Yusenko [13] ✉

Studies of high-entropy materials contribute to various fields of science and reveal ever more exciting properties of applied interest. Here, we perform a study of the resistance of a Cantor alloy (CoCrFeNiMn) to hydrogen through high-pressure experiments at elevated temperatures by X-ray and neutron time-of-flight experiments and ab initio calculations. We report formation of an fcc hydride based on the Cantor alloy composition. We also provide its characterization, including an estimate of hydrogen content. These findings contribute to the growing body of knowledge on the complex chemistry of high-entropy alloys and high-entropy hydrides.

Multicomponent metallic alloys have long been at the forefront of materials science. They play a crucial role in modern life, powering innovations across various technological fields, including the hydrogen economy. Metal hydrides, particularly binary and ternary systems, have been widely explored as promising materials for hydrogen storage in fuel cells[1,2]. However, beyond such and similar simpler systems, research of more complex systems remains relatively scarce. Recently, quaternary body-centred cubic (bcc) AlTiVCr, TiVCrNb, TiZrNbTa, and TiVZrNb systems were characterized under compression in the presence of $H_2$[3,4]. However, despite certain progress, our understanding of ternary and higher-order high-entropy alloys (HEAs), particularly their structural behavior and hydrogen absorption properties, remains

limited. This emerging field, at the intersection of chemistry, physics, and materials science, still holds significant potential for ground-breaking discoveries and applications. In comparison to conventional materials (e.g., 316 L steel and others), the high-entropy counterparts offer a broader range of applications not only because of their attractive mechanical, corrosion-resistant, and other properties, but also due to the large multicomponent phase space that provides greater freedom to tailor these properties.

HEAs represent a relatively new class of multicomponent alloys. They have simple crystal structures and an intrinsic high chemical disorder. Their number of applications is constantly growing. They were initially developed as materials for structural applications[5,6]. At

[1]Deutsches Elektronen-Synchrotron DESY, Hamburg, Germany. [2]Institute for Inorganic and Analytical Chemistry, Goethe University Frankfurt, Frankfurt am Main, Germany. [3]Department of Chemistry - Ångström Laboratory, Uppsala University, Uppsala, Sweden. [4]Institute for Materials Science, University of Stuttgart, Stuttgart, Germany. [5]Department for Computational Materials Design, Max Planck Institute for Sustainable Materials GmbH, Düsseldorf, Germany. [6]Interdisciplinary Centre for Advanced Materials Simulation (ICAMS), Ruhr-Universität Bochum, Bochum, Germany. [7] J-PARC Center, Japan Atomic Energy Agency, 2-4 Shirakata, Tokai-mura, Ibaraki, Japan. [8]Leipzig University, Faculty of Chemistry, Institute of Inorganic Chemistry and Crystallography, Leipzig, Germany. [9]Powder Materials Division, Korea Institute of Materials Science, Changwon, South Korea. [10]Department of Materials Science and Chemical Engineering, Hanyang University ERICA, Ansan, South Korea. [11]ESRF – The European Synchrotron, Grenoble, France. [12]Institute of Materials Physics, University of Münster, Münster, Germany. [13]Institute of Geology, Mineralogy and Geophysics, Faculty of Geosciences, Ruhr-University Bochum, Bochum, Germany. [14]Present address: Beijing Synchrotron Radiation Facility (BSRF), Institute of High Energy Physics, Chinese Academy of Sciences, Beijing, China. ✉e-mail: konstantin.glazyrin@desy.de; kristina.spektor@desy.de; kirill.yusenko@ruhr-uni-bochum.de

the same time, high-entropy materials were also intensively studied as corrosion-resistant, catalytical and energy-related materials[7,8]. The topics of energy storage and conversion are also given due attention in the literature[9]. Particular interest has been paid to the preparation of multicomponent metallic hydrides and characterization of their fundamental properties.

Formerly, bcc-structured HEAs were under the focus of studies concerning the design of materials with a high hydrogen-to-metal ratio. For example, refractory HEA bcc–TiVZrNbHf can absorb much higher amounts of hydrogen than its individual components and reach an enormous hydrogen-to-metal ratio of 2.5[10]. Such a high hydrogen content has never been observed in binary hydrides composed solely of transition metals, considering reasonably moderate synthesis conditions (e.g., 400 °C and ~20 bar $H_2$ pressure). This can be explained by the micro-lattice strains in the alloy, which makes it favourable for hydrogen to occupy both tetrahedral and octahedral interstitial sites within the structure. This observation led to the discovery of several Ti and Mg-based HEAs with high hydrogen uptake, illustrating the HEAs' potential as hydrogen storage materials. The uptake will depend on composition and structure of HEAs. Considering the non-hydrogenated, pure HEAs, here we cite body-centred cubic (bcc), face-centred cubic (fcc), and hexagonal close-packed (hcp) HEAs as the most conventional[11]. The high local crystallographic micro-strain, inherent for all HEAs, may either stimulate the formation of alloys with significant hydrogen uptake in their structure or inhibit hydrogen absorption[12]. Still, despite their multifaceted properties and applications, the reactivity of only a few HEAs has been characterised with respect to their interaction with $H_2$.

The fcc-CoCrFeNiMn alloy, also widely known in the community as the Cantor alloy[13], is among the most studied HEAs, yet many aspects of these materials remain to be characterised, including their reactions with hydrogen as well as potential pathways for such reactions. In our previous study we discovered the surprising stability of the Cantor alloy against hydrogen uptake at room temperature and high-pressure conditions[12]. The Cantor alloy remains extremely resilient to $H_2$ upon compression up to 40 GPa. According to our previous observations, the measured volume per metal atom decreases consistently with increasing pressure and showed minimal deviations from the compression observed in inert pressure-transmitting media, such as helium or neon. Although the final properties depend on specific operation conditions, our observations are placing the Cantor alloy among the most resistant materials with the highest resistance to hydrogen absorption on par, if not better, to the leading Cu–Be alloys and steels. Nevertheless, the challenge of converting the material into a hydride remains and warrants further investigation. Addressing this challenge is crucial for advancing our understanding of HEAs and their potential applications.

In the present work, we experimentally evaluate the impact of temperature (T) and pressure (P) on the P–T–x phase space, where x denotes the H-content in the pseudo-binary system $(CoCrFeNiMn)H_x$ if we simplify the concept of multicomponent high-entropy alloy hydride. We investigate compression of the Cantor alloy in the presence of pure hydrogen with a diamond anvil cell (DAC) and hydrogen formed from ammonia borane precursor with a Large Volume Press (LVP) under high-pressure and high-temperature conditions. We show the formation of an fcc-structured stoichiometric high-entropy hydride (CoCrFeNiMn)H based on the Cantor alloy. We support our experimental results with density functional theory (DFT) calculations, which reinforce our observations and offer additional insight. The calculations suggest thermodynamic stability of fcc-structured hydride, with H occupying octahedral sites.

## Results And Discussion

Upon compression at room temperature, the fcc–CoCrFeNiMn gradually goes through fcc→hcp structural transformation. The transition typically occurs above 7 GPa and depends on various factors, including the pressure-transmitting medium, state of the sample (sintered grains or loose powders) and the degree of hydrostaticity during compression[12,14–16]. For further details, including the observation of phase transition hysteresis and other relevant findings, the reader is referred to the original publications.

The hcp phase of the alloy is formed above 14 GPa and can be recovered as a metastable product at ambient pressure[15,16]. Heating of metastable hcp–CoCrFeNiMn at ambient pressure results in a full recovery of the initial fcc structure. Experimental observations suggest that neither fcc– nor hcp–CoCrFeNiMn absorb significant amounts of hydrogen at room temperature upon compression up to 40 GPa[12].

We conducted several diamond anvil cell (DAC) and large volume press (LVP) experiments under compression and elevated temperatures to enhance the kinetics that may hinder hydride formation at room temperature. Our first DAC attempt (Fig. 1, experiment DAC01; see Methods) showed a formation of fcc hydride in a heating-stimulated reaction between hcp–CoCrFeNiMn and $H_2$ serving as a reagent and a pressure-transmitting medium. The sample was pre-compressed to 40 GPa with a DAC and then heated at 573 K in a vacuum oven. Similar to previous studies[12], our analysis and derived conclusions are based on a comparison between the observed atomic volume, V/Z (here, V to represent the volume of a unit cell and Z to denote the number of formula units per unit cell), and the values measured for unreacted material, compressed either in a hydrostatic pressure medium or in $H_2$, and measured using X-ray diffraction (XRD). Our observations of hydride formation are also supported by the increased peak width of the sample diffraction signal providing additional evidence of hydrogenation. Despite a decrease in sample chamber pressure by heating a DAC in the vacuum oven, we consider our observation of the fcc phase quenched to 14.2(1) GPa and 293 K as the first proof of principle synthesis of high-entropy hydride (HEH) based on the Cantor alloy. Due to several limitations, deriving additional details regarding hydride formation was not possible in this experiment. However, our further studies using DAC and LVP clarify several important factors, including the P–T range of hydride formation and viable synthesis pathways, as described below.

Next, we confirmed the reactivity of the Cantor alloy with hydrogen at low pressures in our LVP syntheses (Experiments LVP01 and LVP02; see Methods below). The data measured from the product of a reaction between fcc-CoCrFeNiMn alloy and $H_2$ (released by decomposition of $NH_3BH_3$ in an LVP) exhibited signs of hydride formation similar to our DAC experiments. The precursor $NH_3BH_3$, used as a hydrogen source, began producing hydrogen above approximately 600 K. Importantly, no free hydrogen was present prior to the start of $NH_3BH_3$ decomposition. Although the hydrogen chemical potential cannot be easily extracted directly from DAC/LVP experiments, the strong confinement of hydrogen at multi-gigapascal pressures typically leads to increase of hydrogen fugacity[17], providing the thermodynamic driving force for hydride formation.

During our experiments, we observed an increase in atomic volume per metal because of fcc-phase of the alloy hydrogenation (Fig. 2). At elevated temperatures, the Cantor alloy rapidly absorbed hydrogen, forming a fcc-hydride phase with an increased atomic volume per metal. The measured atomic volume gradually increased with the decomposition of the hydrogen source, eventually reaching a value comparable to that observed in our DAC experiments. During the subsequent cooling of LVP assemblies, we recorded a hysteresis of atomic volume confirming presence of hydride phase based on Cantor alloy at the P–T conditions indicated in the Fig. 2. Data from our LVP synthesis confirms the potential to synthesize a Cantor alloy-based hydride at low pressure regime. Further details, including energy-dispersive XRD (ED–XRD) data compilations showing the hydrogen uptake process as a function of temperature and time during LVP01 experiments, are provided in SI. The LVP experiments were conducted

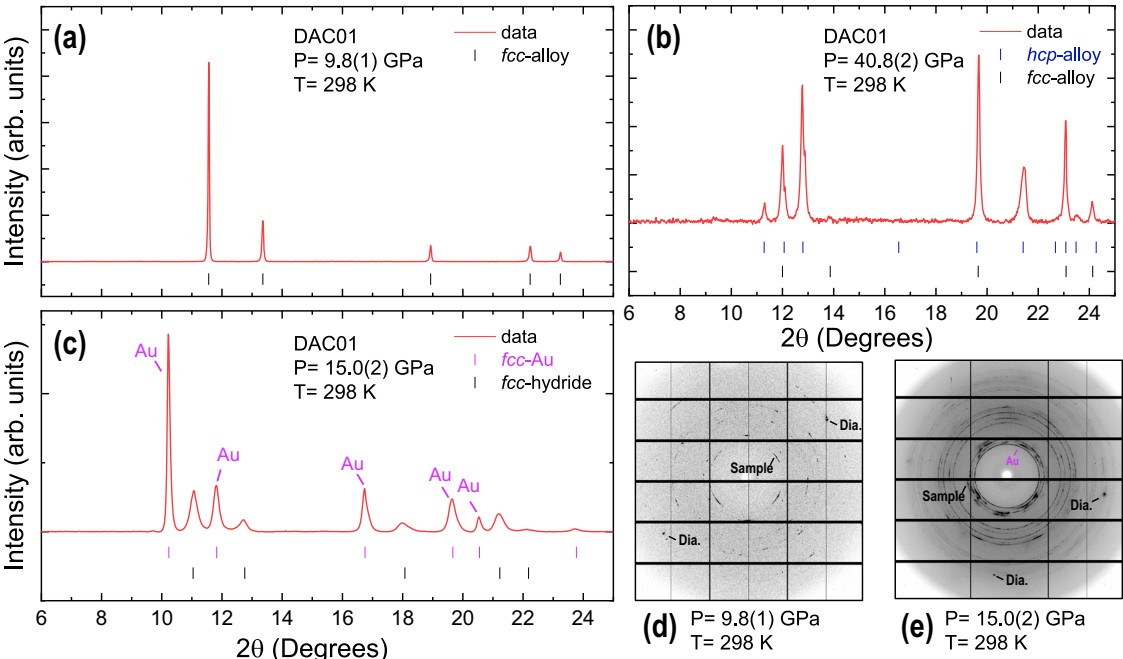

**Fig. 1 | Selected powder X-ray diffraction patterns illustrating hydride formation.** The patterns were collected at the ID15B beamline, ESRF, with a wavelength of λ = 0.4100 Å (experiment DAC01) at ambient temperature. Phases including fcc-alloy, its hydride are indexed for clarity as well as contributions from the hcp-alloy and Au (pressure marker). The corresponding tick marks indicating different phases are shown under the patterns, red lines represent 1D patterns of the raw experimental data. Panel (**a**) shows the sample under compression in $H_2$ at ambient temperature. A DAC containing the sample was heated to ~573 K at an initial sample chamber pressure of ~40.8(2) GPa, corresponding to (**b**). Then the DAC was cooled down to ambient temperature. Upon the heating-cooling cycle, the pressure in the sample chamber decreased, resulting from relaxation of the DAC within the larger mechanical assembly. The resulting pattern is shown in (**c**). Comparison of the fcc-alloy signal in (**a**) with the fcc-hydride signal in (**c**) clearly indicates a larger unit cell volume for the latter. Here, a shift of diffraction peaks to lower 2θ directly corresponds to an increase in the lattice parameter via Bragg's law. Notably, the peaks attributed to the fcc-hydride phase exhibit broadening similar to the effect seen in pure metals. Panels (**d**, **e**) show 2D diffractograms corresponding to the panels (a, c), respectively, and show diffraction patterns collected using micro-beam.

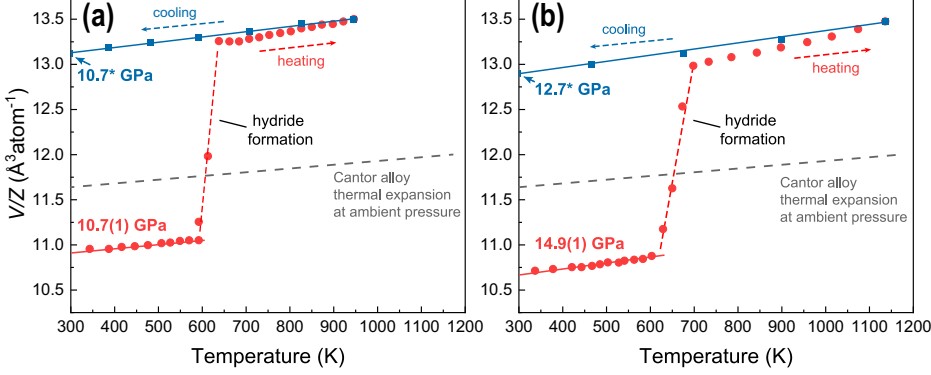

**Fig. 2 | Synthesis of hydride based on the Cantor alloy at different pressures using LVP.** Data shown in panels (**a**, **b**) were collected at the energy dispersive beamline P61B, PETRA-III (experiment LVP01). We indicate the change of measured atomic volume (V/Z). Blue, red and grey lines were fitted linearly. The starting and the final pressures are indicated to the left of the panels using the red and the cyan colors corresponding to the heating and cooling, respectively. Using the grey dashed line, we indicate thermal expansion of the Cantor alloy reported previously for ambient pressure (Ahmad et al. 2017). Hydride is formed on expense of $NH_3BH_3$ decomposition. Lines are used as visual guides for clarity. Additional information is presented in the supplementary material. Considering the V/Z values, the datapoint error bars are of the same size as the symbols or below. Due to experimental limitations pressures were not measured along the cooling path. However, we note that during cooling, the force applied to the large volume press anvils remained constant. Thus, based on our experience, we expect the pressure values indicated by (*) symbols to represent higher estimates of the sample pressure.

at relatively high temperatures due to the limitations associated with the specifics of the $NH_3BH_3$ decomposition process, which hindered our ability to draw definitive conclusions about the temperature sensitivity of the alloy's hydrogenation behavior. A resistive-heating DAC experiment using pure hydrogen as the precursor helped us shed light on the bigger picture.

Below we report an extension of our DAC work providing an insight of Cantor alloy corrosion resistance vs. $H_2$ under compression at 378 K (Experiment DAC02; see Methods). Supplementary to ambient temperature data[12], we note that Cantor alloy shows a remarkable resilience to hydrogen also during resistive heating (Fig. 3). However, we also note that compression with simultaneous heating forces HEH formation under compression starting from ~16 GPa and completing at higher pressures. In supplementary we present additional information.

Within the Fig. 4 we sum up our observations for different experiments. We see a good agreement of fcc-Cantor alloy

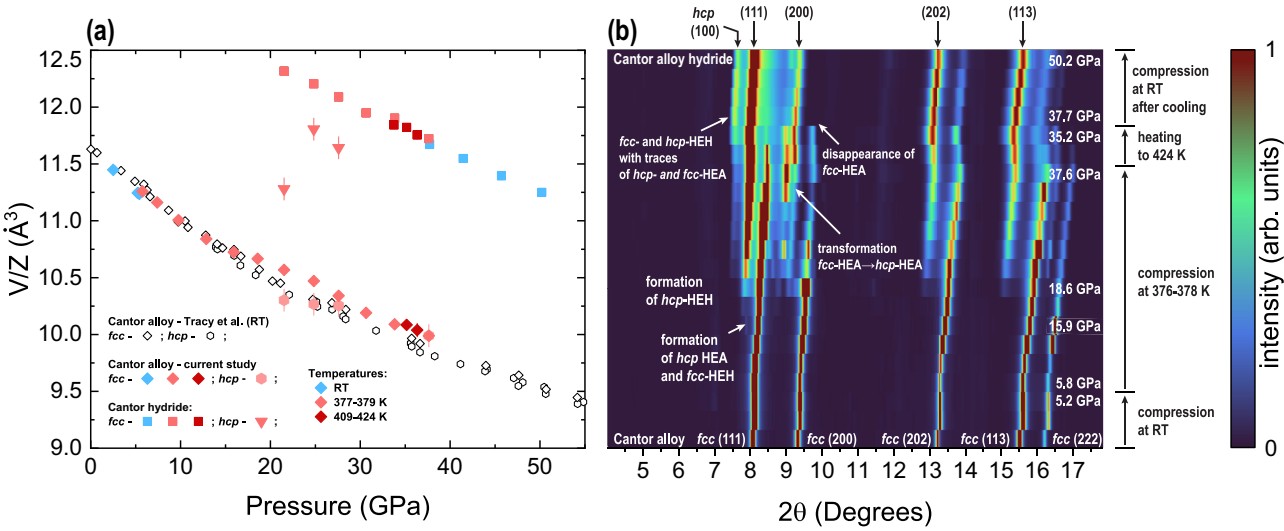

**Fig. 3 | Formation of hydrides from fcc phase of the Cantor alloy under compression at elevated temperatures.** Powder XRD data were collected at P02.2, PETRA III using λ = 0.2907 Å (experiment DAC02). Panel (**a**) shows the variation of V/Z for the for the identified phases formed upon compression, resistive heating and reaction of the Cantor alloy with hydrogen. Note that due to diffraction peak overlap we could not unambiguously resolve unit cell volume of hcp–HEA and hcp–HEH for a number of points. For comparison, we show room temperature (RT) (RT) compressibility similar to our previously published data[15]. Unless indicated, error bars for the datapoints are of the same size as the symbols or below. Panel (**b**) illustrates a heatmap composed from individual experimental diffraction patterns. The most important processes are indicated, including formation of fcc– and hcp-HEH based on Cantor alloy. Transformation of fcc alloy into hcp phase happens over an appreciable range of pressures. In addition, we observe domination of HEH with respect to HEA polymorphs at highest pressures indicated to the right of the panel. Diffraction data suggests relative abundance of fcc–HEH with respect to hcp–HEH.

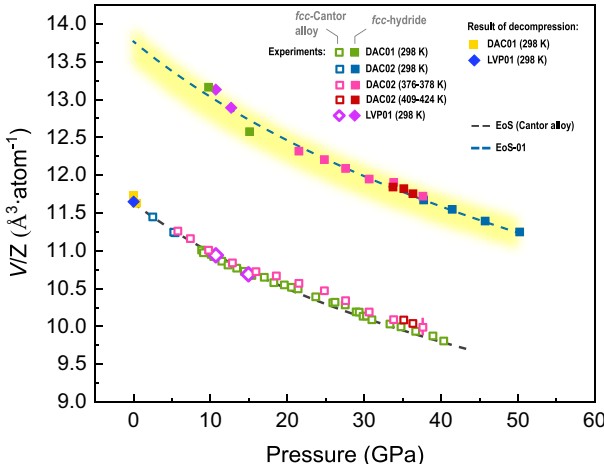

**Fig. 4 | Atomic volume of the fcc-structured Cantor alloy and the fcc-hydride synthesized through different pathways.** We indicate different experiments and their conditions. Using the grey dashed line, we indicate equation of state (EoS) of the Cantor alloy reported[15]. The yellow shading represents our estimates of the hydride volume excess per metal atom, assuming a single hydrogen atom per metal atom (see the text for discussion). The dashed blue line corresponds to the 2nd Birch-Murnaghan EoS estimate, calculated based on the hydride phase datapoints collected at ambient temperature. We report $K_0 = 164(10)$ GPa, $V_0/Z = 13.8(1)$ Å³/atom and K' = 4 (fixed) for the blue dashed line (EoS-01). Unless indicated, error bars for the datapoints are of the same size as the symbols or below. The pressures for data attributed to the fcc-hydride obtained in LVP01 represent upper estimates. These points were not considered during EoS-01 calculation.

compression curve measured by Tracy et al. [15] obtained at ambient temperature with our data. A slight deviation from the compression curve is seen for the experiment DAC02 for pressures above 30 GPa. It could be attributed to the imprecision of data analysis associated with decreasing signal to noise ratio during the hydride formation at 376–378 K. The high-pressure data for the fcc-hydride phase can be

described using the formalism of Birch–Murnaghan (BM) equation of state (EoS)[18], and we indicate the calculated values for $K_0$ (bulk modulus at ambient pressure), $V_0/Z$ (atomic volume at ambient pressure), and K' (pressure derivative of the bulk modulus at ambient pressure) for the 3nd BM EoS within the figure caption.

The yellow shading in Fig. 4 highlights our estimated hydride volume, based on the assumption of an hydrogen-to-metal ratio of 1:1. Considering transition metals, hydride composition could be estimated using an assessment of atomic volume of interstitial hydrogen in individual binary hydrides as described by Somenkov et al. [19]. Following the same approach, we can assume the interstitial hydrogen as highly incompressible. Indeed, the largest contribution to the material's unit lattice compressibility comes from the electron density of the metal atoms.

Although the information on high-entropy alloys and their reactions with hydrogen is limited, there is abundant information regarding hydrides of pure elements like Mn, Ni, and Fe. The literature review shows that they form MH hydrides at relatively low pressures of < 1.5, < 2, and <4.5 GPa[20], respectively. For $FeH_x$ the excess of atomic volume corresponding to hydrogen atom insertion ΔV(H) has been estimated as 2.36 Å³/atom[21–23]. For $CoH_x$ and $MnH_x$, the corresponding excess volumes were estimated to be lower: 1.90 and 1.85 Å³/atom, respectively. Considering the latter data and the mentioned below, we kindly guide to the reported values in the recent book[20] and the references therein. In all cases, a higher hydrogen solubility was observed in fcc-structured phases. Considering other possible structures, bcc-structured metals typically do not absorb much hydrogen below their pressure-induced transition into fcc-structured polymorphs. Co and Cr may start forming MH hydrides at relatively low pressures, depending on temperature[20,24,25]. Several ternary and four-component Fe-based hydrides with Ni and Mn, such as fcc–$(Fe_{0.65}Mn_{0.29}Ni_{0.06})$ $H_{0.95}$ with ΔV(H) = 2.09 Å³/atom as well as fcc–$Ni_{0.8}Fe_{0.2}H$ (with ΔV(H) = 2.09 Å³/atom) were also structurally studied. Based on literature data, one can conclude that the magnitude of ΔV(H) in fcc-structured hydrides may vary at ambient temperature between 1.85 and 2.36 Å³/atom. After considering the equimolar Cantor alloy

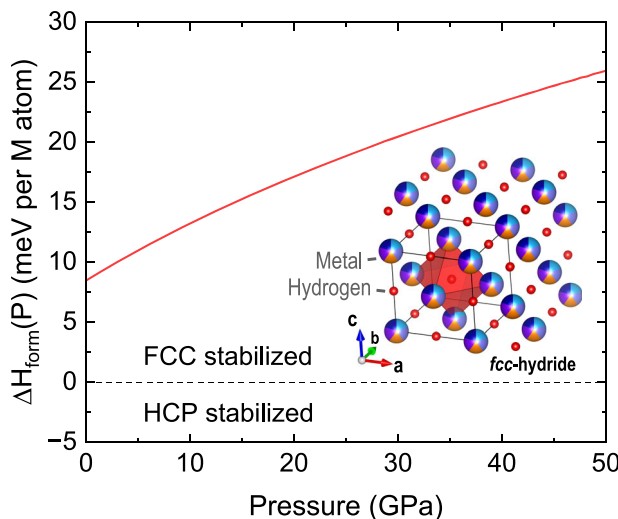

**Fig. 5 | Results of DFT calculations exploring hydride polymorph stability.** Pressure dependence of the computed formation energy difference of the hcp- and fcc-structured octahedral hydrides. Calculations performed for the case of 0 K emphasize stability of fcc over hcp. Inset shows a sketch of fcc-hydride with metal and hydrogen atoms indicated. We use the red octahedron to highlight a single octahedral site.

composition, the average atomic volume is estimated to be approximately ~2.1 Å³/atom. The yellow shading shown in Fig. 4 was calculated by considering the EoS reported previously[15], including the corresponding values of $K_0$ and $K'$, with $V_0$ of hydride expanded in comparison to the hydrogen-free compound by a contribution of a single H atom using the values within the indicated range. The lower and the upper boundaries of the shading correspond to ΔV(H) of 1.9 and 2.2 Å³/atom, respectively. Given the complexity of the Cantor alloy composition and its hydride, we observe a very good agreement between our simple model and the data. Based on the hypothesis of 1:1 H:M hydride formation in materials composed of 3 d transition metals, it is logical to assume that hydrogen occupies the octahedral interstitial sites, but the latter requires further theoretical and experimental evidence. To gain deeper insight, we performed several DFT calculations.

In elemental transition metals, hydrogen occupying the octahedral interstitial site typically has a lower energy than hydrogen in the tetrahedral site. Nevertheless, the type of site H will occupy will depend on specific conditions (e.g., specific composition, as well as structure of the host metal lattice). To determine whether the dilute-limit findings for pure metals also imply a stabilization of octahedrally filled hydrides in the Cantor alloy, we first assessed the energetic preference between octahedrally and tetrahedrally occupied fcc-structured hydrides. This was done by evaluating the formation enthalpy at zero pressure for both octahedral and tetrahedral types of hydrides. The formation energies per metal atom M were defined as:

$$\Delta H_{form}(MH_x) = \frac{1}{n_M}\left(H_{sc}(MH_x) - H_{sc}(M) - \frac{n_H}{2}\mu_{H_2}\right) \quad (1)$$

Here, $H_{sc}(MH_x)$ and $H_{sc}(M)$ are the zero-pressure enthalpies of the metal-hydrides and metal supercells, $\mu_{H_2}$ is the chemical potential of a $H_2$ molecule, and $n_M$ and $n_H$ are the numbers of metal and H atoms in the metal-hydride supercells, respectively. For filling all octahedral sites with 54 H atoms, x equals 1. For filling all tetrahedral sites with 108 H atoms, x equals 2.

For the zero-pressure formation energy difference of the completely filled tetrahedral and octahedral site scenarios, we obtained a value of −0.199 eV per metal atom in favor of the occupation of octahedral sites, which agrees with the preference of H filling the

octahedral site in the constitutive pure metals. This is also consistent with our experimental finding of an hydrogen-to-metal ratio of 1:1. Notably, the derived atomic volume attributed to hydrogen also matched the expected value of 2.13 Å³/atom.

To interpret the experimentally observed stability of the fcc vs. hcp-structured hydride under pressure (P), we performed additional calculations for the hcp-structured octahedral hydride. For this purpose, we consider the formation enthalpy differences:

$$\Delta H_{form\ MH}(P) = H_{form\ MH}^{hcp}(P) - H_{form\ MH}^{fcc}(P) \quad (2)$$

As we show in Fig. 5, $\Delta H_{form\ MH}(P)$ has a positive value at ambient pressure. It also demonstrates a positive trend as a function of pressure. Our results suggest that the known pressure-induced stabilization of the hcp-phase in the absence of hydrogen is reversed in the presence of hydrogen, favoring the formation of fcc-structured high-entropy hydrides. According to our DFT calculations, the stability of the fcc-structured polymorph increases relative to the hcp polymorph across the entire experimental pressure range covered in this study. This is also consistent with our observations. Finally, although our DFT calculations indicate a preference for hydrogen to occupy octahedral sites in the dominant fcc structure under ambient conditions, we were unable to extend these calculations to high-pressures. We confirmed the octahedral site preference by performing additional DFT calculations for a half-filled (checkerboard) tetrahedral hydride, demonstrating that the octahedral arrangement remains energetically favored across all pressures (see Supplementary), consistent with recent dilute-limit calculations[25]. Our computational results are also verified by the neutron scattering experiments described below.

In comparison to X-ray diffraction, neutron experiments exhibit significantly higher sensitivity to hydrogen contribution, due to a much larger cross-section (i.e., deuterium in elastic scattering experiments). Neutron scattering allows precise localization of hydrogen, typically introduced as deuterium, within crystal lattices. We conducted an in-situ experiment featuring synthesis of the Cantor alloy deuteride at the PLANET beamline at MLF, J-PARC (experiment LVP02; see Methods). The synthesis pathway was similar to LVP01 but conducted at lower pressures (5.7 GPa) and slightly different temperatures (as shown in Supplementary). Similar to LVP01, the sample was transformed and deuterated upon heating to 850 K yielding direct observation of the fcc-structured deuteride phase. Our data analysis of experimental data confirmed finite occupation of deuterium at octahedral positions of the high-entropy hydride. For example, in the Fig. 6 we show a Rietveld refinement for the experimental point of 5.7 GPa and 850 K. The refinement yields a $R_w$ of 2.76% and goodness-of-fit $\chi^2 = 1.08$. The corresponding unit cell parameter a = 3.7347(1) Å and V/Z = 13.023(1) Å³/atom. The inset in the figure illustrates the octahedral void occupied by deuterium, as revealed by a difference Fourier map during the analysis of collected data. The excess volume per atom ΔV(H) for the data shown in figure was estimated to ~1.5 Å³/atom, which is smaller in comparison to the average value of ~2.1 Å³/atom indicated above. We expect some small difference between the atomic volumes of interstitial deuterium and hydrogen (see the ongoing discussion of ref. 21,23. and the references within). Thus, considering the composition of fcc–(CoCrFeNiMn)$D_x$, we suggest that, while the data Rietveld refinement yields an occupancy factor slightly larger than 1 (fixed at 1.0 during the later refinement stages), the point of 5.7 GPa and 850 K results in a hydrogen concentration slightly lower, conservatively estimated as 0.75(4) in correlation with the measured ΔV(H). This observation matches our results shown in the Fig. 6 indicating variable hydrogen content and release of hydrogen on decompression at ambient temperature.

It is common for H to occupy only an octahedral site. Occupation both in octahedra and tetrahedral sites is limited only to lanthanoids at moderate pressures or elemental metals at ultra-high pressures.

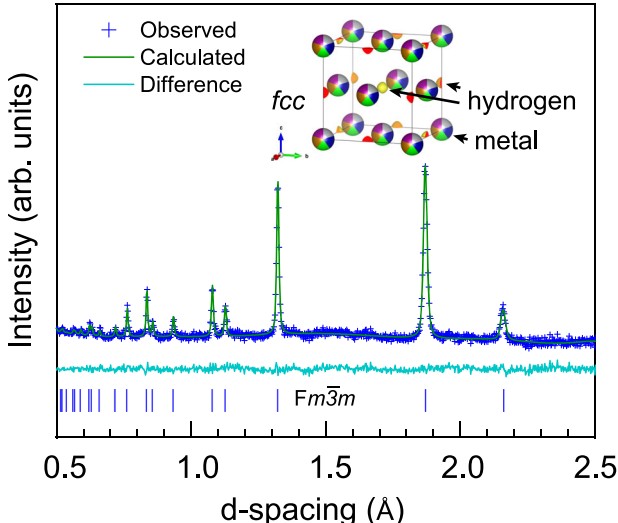

**Fig. 6 | Rietveld refinement of the fcc–(CoCrFeNiMn)$D_x$ phase measured at 5.7 GPa and 850 K.** The studied deuteride was synthesized in situ and measured at the PLANET beamline. The inset illustrates the resolved deuterium position (shown in orange and yellow) within the octahedral interstices of the fcc-metal lattice used in the refinement, as determined from a difference Fourier map. The patterns shown in the figure correspond to observed data (blue cross symbols), while green and light blue lines represent fit of a Rietveld model and the difference between the experimental signal and calculated pattern, respectively. The tick-marks shown at the bottom of the figure indicate the expected fitted peak positions for the fcc model.

However, local distortions of cubic symmetry and partial occupancy of tetrahedral sites should be examined further through precise neutron experiments in a wider pressure range. We also know that bcc-structured alloys may undergo a transformation to a fcc-structured hydride upon hydrogenation with the hydrogen taking over some tetrahedral sites[26].

Before we move to the conclusions, we would like to reiterate the following. In the DAC experiments (DAC01, DAC02), different P–T paths yielded the same results, indicating that hydride formation (including the final composition as reflected in the unit cell volume values) depends mainly on the final pressure–temperature conditions, not on the synthesis pathway. It is important to note that both temperature and pressure play important roles in the formation of the hydride. Indeed, both factors should affect hydrogen fugacity, and our compression at 100 °C shows that hydride forms at moderate pressure. In contrast, at room temperature, the hydride does not form even at the highest pressures applied in our study, whereas pure iron, for example, forms a hydride at ~3–4 GPa at RT.

In addition, in the LVP experiments (LVP01, LVP02), the main difference was the probe: LVP02 used in-situ neutron diffraction, LVP01 used X-rays. Both produced the same hydride phase, confirming that the outcome is independent of the detection method and allowing experimental localization of hydrogen.

Combining DAC and LVP data contributes to redundancy and exploits their complementary strengths: higher pressures in DAC and larger volumes with neutron sensitivity in LVP. The agreement across all four experiments shows that final P–T conditions control hydride formation, while differences in synthesis pathways or techniques are secondary.

The literature review, including refs. 27,28, highlights the complexity of studying the resistance of Cantor alloys to the multifaceted phenomenon of corrosion, including their permeability to hydrogen. This complexity may lead to interpretational inaccuracies, as individual studies often provide only a single perspective on a broader challenge. It should be emphasized that the answers in this field are far

from straightforward. It is worth noting that some reports lack a thorough chemical analysis of the constituent elements, which is crucial for developing a comprehensive understanding. Moreover, results obtained from conventional methods (such as electrochemical and autoclave experiments) are not always directly comparable to each other or to the actual conditions under which practical components operate. At the same time, our results contribute an additional and important aspect to the overall multivector characterization by examining bulk hydrogen saturation in the material at specific temperatures, a perspective missing from conventional studies, which either typically employ higher temperatures to compensate for time limitations or from electrochemical approaches, which operate on an entirely different scale than methods described in our paper.

Cantor alloy, fcc–CoCrFeNiMn, does not absorb hydrogen at room temperature, even at relatively high pressure (above 40 GPa), which hints to potential applications as hydrogen-resistant material[12]. Our new data confirms resilience of Cantor alloy to hydrogenation even at elevated temperatures. At a temperature of 378 K (slightly above 100 °C), which is highly relevant for industrial applications, the high-entropy hydride of the alloy forms at pressures above 15.9(2) GPa, which exceed significantly those encountered in practical applications.

Although high-entropy hydrides still represent an under-investigated class of compounds, our study provides valuable insight. In the current study employing the DAC and LVP high pressure techniques, we investigated formation of a hydride based on the well-studied Cantor alloy and demonstrated various paths of synthesis. Although the hydride could not be quenched to ambient conditions, we were able to characterize its structure and details of hydrogenation process by means of in situ X-ray and neutron experiments in combination with DFT calculations. Among other results, we show the stability of fcc-structured hydride over hcp- counterparts and provide a reasonable estimate for the resulting hydrogen concentrations. The addition of neutron experiments and DFT calculations data provided important insight into atomic-level details, revealing that hydrogen occupies the octahedral sites of the fcc structure. These results suggest that the presence of hydrogen at tetrahedral sites is unlikely under the studied conditions. This is important for understanding the influence of intrinsic strains caused by electronic density variations associated with the inherent atomic disorder of high-entropy alloys and their hydrides. Our results also suggest that under low-pressure and high-temperature conditions, the hydrogen concentration can be less than 1, providing further evidence for the Cantor alloy's toughness against hydrogenation. Upon decompression, hydrogen is released from the crystal lattice. At the same time, at pressures beyond the P-T range of relevance for practical applications, we see moderate hydrogen uptake of around 1 hydrogen per 1 metal atom.

Our study adds another piece to the puzzle of high-entropy alloys and their reactions with hydrogen, formation of high-entropy hydrides. We hope that our results will attract additional attention to the underexplored field of high-entropy solids and their potential applications in the fields of hydrogen economy and beyond.

## Methods

The fcc-structured CoCrFeMnNi HEA powder was produced at the Korea Institute of Materials Science (KIMS) using a vacuum induction gas atomizer (VIGA, HERMIGA 100/25, PSI, UK). It is the same as used for additive manufacturing by laser power bed fusion[29,30]. A cast and homogenized ingot with the sought equiatomic composition was first produced and then gas-atomized at 1580 °C in an argon atmosphere. An inspection of the resulting CoCrFeMnNi material revealed small-grain polycrystals with dense microstructure, nearly equiatomic composition, and a homogeneous distribution of the alloying elements[31,32]. Powder X-ray diffraction data was collected at various P–T conditions and synchrotron facility beamlines either focused on DAC or LVP work as described below.

Room temperature compressibility curve for fcc–CoCrFeMnNi Cantor alloy in $H_2$ as pressure transmitting medium and reactive gas was collected at the ID15B beamline up to 40 GPa (ESRF/EBS, $\lambda = 0.4100$ Å, Dectris EIGER2 CdTe 9 M hybrid pixel detector, beam size $2(v) \times 2(h)$ μm²). The two 10–15 μm balls of sample material were loaded in a symmetric DAC with conically supported Boehler Almax anvils (300 μm culet size)[33]. Small pieces of gold powder placed near the sample were used to calibrate pressure[34]. After reaching the target pressure of 40 GPa, DAC was externally heated in a vacuum oven (VT6025; Kendro Laboratory Products) to 573 K (200 °C). We kept the DAC at same temperature overnight (~8 h) and let the DAC together with sample material and hydrogen surrounding it to cool down naturally to room temperature. After heating, the pressure inside the sample chamber dropped to ~15 GPa due to a relaxation of the DAC as a mechanical assembly supplemented with a reaction of Re gasket material with $H_2$. Within the manuscript we indicate the experimental session as DAC01.

Further high-pressure, high-temperature experiments in $H_2$ as pressure transmitting medium and reactive gas in resistive heated DACs were conducted at the P02.2 beamline[35] of PETRA-III, DESY up to 55 GPa ($\lambda = 0.2907$ Å, Perkin Elmer XRD1621 detector, beam size $3(v) \times 8(h)$ μm², FWHM). The samples were loaded in a symmetric DAC equipped with Boehler Almax anvils (300 μm culet size). Similar to other DAC experiments, gold powder placed near the sample was used as pressure calibrant (Supplementary Fig. S3)[34]. The sample was compressed to 6 GPa and then heated to 376–378 K using a whole cell heater available at the beamline. Temperature was controlled using a type–K thermocouple mounted at the back of a diamond anvil of a DAC. Following the initial heating at low pressures, sample material was further compressed to 36 GPa and finally heated to 423 K. At each point, the sample was equilibrated for a minimum of 5–10 min (Supplementary Figs. S1 and S2). After reaching 423 K, the sample was cooled down to room temperature and compressed to a final pressure of 48 GPa. Here and below we indicate the experimental session as DAC02.

Considering our X-ray diffraction data collected from DACs, 2D diffraction data were integrated into 1D profiles using DIOPTAS[36]. The unit cell parameters, the background and the line-profile parameters were refined simultaneously using TOPAS 6.0[37] or GSAS-II[38] software. The P–V data were fitted using EoSFit7[39,40]. The error bars defined in the Figs. 2, 3 & 4 were obtained from the analysis software. Typically, they correspond to the confidence interval of 95 % - 2σ.

High-temperature curves under pressure for fcc–CoCrFeMnNi Cantor alloy in $H_2$ fluid were collected in the Hall-type 6-ram large volume multi-anvil press Aster–15 (MAVO press LPQ6-1500-100) installed at the P61B energy-dispersive beamline[41] of PETRA-III, DESY. 6 mg of fcc–CoCrFeMnNi Cantor alloy powder was pressed into a pellet (1.2 mm OD, 0.75 mm height) and sealed inside a NaCl capsule (3 mm OD, 3.4–3.7 mm height) along with two pellets of $NH_3BH_3$ (ammonia borane, 1.2 mm OD, 0.75 mm height). $NH_3BH_3$ (Sigma Aldrich, 97%) was employed and as a hydrogen source (see also[12] where we report and discuss room temperature compressibility data of the same experiment), providing ~2.5 times molar excess of $H_2$ precursor with respect to the metals of Cantor alloy. $NH_3BH_3$ has a well-defined decomposition behavior at high pressures and produces chemically inert BN as residual[42]. The sample capsule preparation was handled in an Ar-filled glove box. Samples were compressed to target pressures using 14/7 multianvil assemblies[43]. Pressures were estimated using the NaCl equation of state[44]. Temperature was evaluated from power – T calibration curves obtained by reproducing in situ runs offline using analogous 14/7 setups with central type-C thermocouples. The sample was compressed up to 11 or 15 GPa and carefully heated to decompose $NH_3BH_3$ and form $H_2$. On compression, heating, cooling, and decompression, PXRD data were simultaneously collected similar to our previous study[12]. Energy-dispersive X-ray diffraction patterns were visualized and inspected using PDIndexer software tool[45]. Further details, including plots showing compiled EDXRD data, are provided in

SI (Supplementary Figs. S4 and S5). Within the manuscript we indicate this experimental session as LVP01.

In-situ high pressure high-temperature neutron powder diffraction patterns were obtained using the ATSUHIME 6-axis press[46] installed at the PLANET beamline at the Materials and Life Science Experimental Facility at the J-PARC, Tokai, Japan[47]. The compression geometry used was MA6-6 with TEL10 mm Ni-binded WC second-stage anvils. The powder sample, 0.244 g of fcc–CoCrFeMnNi, was pressed into a pellet (3.5 mm OD, 4 mm height) and enclosed in a NaCl capsule (5.5 mm OD, 9 mm height) along with two pellets (total 0.0375 g) of fully deuterated ammonia borane $ND_3BD_3$ (3.5 mm OD, 2 mm height each) within a graphite furnace analogously to the synchrotron experiments at P61B (LVP01). The synthesis of $ND_3BD_3$ was done according to published procedure with one additional deuteration cycle[48]. All steps were performed as inert as possible. Ammonium formate, $NH_4(HCOO)$, was dried under vacuum for 12 h. A 1:1 reaction mixture of ammonium formate and $NaBD_4$ in dried THF was sonicated for 30 min (275 W). After inert filtration, THF was removed under vacuum, and the remaining $NH_3BD_3$ was deuterated with $D_2O$ in four deuteration cycles. The product was dried overnight under vacuum and characterized via Raman spectroscopy. Unlike the synchrotron experiment, which employed the Kawai-type 6-8 assembly with octahedral sample geometry, the six second-stage anvils here accommodate a cubic solid pressure-transmitting medium, allowing larger sample volumes at the expense of pressure reach. The sample was compressed in ca. 0.5 GPa steps to 5.7 GPa and subsequently heated to 850 K in 100 K steps. Time-of-flight diffraction data were collected using $^3$He position sensitive detectors on each step during compression and heating in the d-range 0.2–4.2 Å. Data from vanadium and empty cell assembly and instrument were collected for sample data correction. Temperature was controlled based on the pre-calibrated heater power relationship, and pressure was estimated from the lattice parameter of NaCl[49]. Rietveld refinement and difference Fourier map calculations were performed using the GSAS-II software[38]. Within the manuscript we refer to this experimental set as LVP02.

In the DFT calculations, the CrMnFeCoNi HEA was modelled based on the supercell approach. We utilized 54-atom fcc and hcp supercells with identical simulation-cell shapes as proposed by Ikeda et al.[50] with <111> direction along the z-axis. These supercells have six layers, and each layer consists of nine atoms. The ideal mixing of the elements in CrMnFeCoNi was approximated based on the method of special quasirandom structures (SQSs)[51]. The first and the second nearest-neighbor pairs were optimized to be close to the ideal mixing state. A single five-component SQS configuration for each of the fcc and hcp phase was constructed. Note that the 54-atom SQSs have a composition ratio of 11:11:11:11:10, which slightly deviates from the equiatomic, and hence, the energies of equiatomic CrMnFeCoNi were evaluated by taking the average over five supercell models, commuting the elements. Two hydrides were simulated, $(CrMnFeCoNi)H_2$ and $(CrMnFeCoNi)H$, filling all available 108 tetrahedral sites or all 54 octahedral sites with hydrogen, respectively.

The electronic structure calculations were carried out using the plane-wave basis projector augmented wave (PAW) method[52] and using the generalized gradient approximation of the Perdew-Burke-Ernzerhof[53] as implemented in the VASP code[54–56]. The plane-wave cut-off energy was set to 400 eV. The Brillouin zones were sampled by a Γ-centred $4 \times 4 \times 4$ k-point mesh for the 54-atom supercell models, and the Methfessel-Paxton scheme[57] was employed with the smearing width of 0.1 eV. The 3d4s orbitals of Cr, Mn, Fe, Co, and Ni were considered as valence states. The total energy was minimized until it converges within $10^{-3}$ eV per simulation cell for each ionic step (see also Supplementary Fig. S7). Ionic relaxations were performed until the residual forces became less than $5 \times 10^{-2}$ eV/Å. The shapes of the supercells were kept fixed during calculations. The ideal c/a ratio was applied for the hcp phase calculation. All calculations were performed

by considering spin polarization; all the magnetic moments on Cr and Mn were initially set to be antiparallel to those on Fe, Co, and Ni (see also Supplementary Fig. S6). A set of at least 10 energy-volume points has been computed, fitted to the Vinet equation of states to obtain the equilibrium volume for each phase.

Large Language Models (LLMs), including ChatGPT (OpenAI™) and Gemini (Google™), were utilized for English grammar and vocabulary checking during the writing stage of this manuscript. The use of these LLMs was limited to language refinement and did not influence any results, interpretations, or conclusions presented in the manuscript.

## Data availability
The Source Data underlying the figures of this study are available with the paper. All raw data generated during the current study are available from the corresponding authors upon request as well as can be downloaded from the DARUS host database online under https://doi.org/10.18419/DARUS-5698

## Code availability
Simulation files and atomic coordinates for models are available from the DARUS host database online under https://doi.org/10.18419/DARUS-5698

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

## Acknowledgements

We acknowledge DESY (Hamburg, Germany), a member of the Helmholtz Association HGF, for the provision of experimental facilities. Parts of this research were carried out at beamline P02.2 and P61B (Proposal No. I-20220244) with the support from the Federal Ministry of Education and Research, Germany (BMBF, grants no.: 05K16WC2 & 05K13WC2). We are grateful to Dr. Robert Farla for his support during P61B experiments. K.S. thanks Per Mistenius for skillfully manufacturing the miniature press dies used for sample preparation. We also thank the ID15B beamline at the European Synchrotron Radiation Facility, Grenoble, France, for providing us with the measurement time and technical support (Experiment No MA-5925 with corresponding DOI for collected experimental data 10.15151/ESRF-ES-1437868272) as well as we thank L. Brüning and P. Jurzick (Johann Wolfgang Goethe-Universität Frankfurt am Main) for their help during the experiment as the ESRF. Y.I. is funded by the Deutsche Forschungsgemeinschaft (DFG, German Research Foundation) – 519607530. F.K. and B.G. acknowledge funding from the European Research Council (ERC) under the European Union's Horizon 2020 research and innovation programme (grant agreement No. 865855). F.K. also acknowledges funding through the Heisenberg Programme of the Deutsche Forschungsgemeinschaft (DFG, German Research Foundation) – project number 541649719. Neutron diffraction experiments at the Materials and Life Science Experimental Facility of the J-PARC were performed through the user program (Proposal No. 2023A0078 and 2023B0335). P.H.B.B.C. acknowledges the financial support from the SSF-SwedNESS grant number SNP21-0002. M.B. acknowledges the support of Deutsche Forschungsgemeinschaft (DFG Emmy-Noether Program project BY112/2-1) co-funded by the European Union (ERC, HIPMAT, 101077963). Views and opinions expressed are, however, those of the author(s) only and do not necessarily reflect those of the EU or the ERC. M.B. also acknowledges the support of Johanna-Quandt Young Academy, Loewe Starting Professorship Program of the state of Hesse, as well as of Adolf-Christ Foundation. S.D. acknowledges partial funding from DFG, project number 509804947.

## Author contributions

Konstantin Glazyrin, Kristina Spektor, and Kirill Yusenko conceived the research, developed the methodology, performed experiments, acquired funding, and wrote the original manuscript draft. Fritz Körmann developed the modelling methodology, performed the calculations, acquired funding, and wrote the original manuscript draft. Paulo H.B. Brant Carvalho, Asami Sano-Furukawa, Takanori Hattori, and Martin Sahlberg performed the high-pressure neutron experiments, and experimental analysis, and participated in manuscript review and editing. Maxim Bykov, Weiwei Dong, and Shrikant Bhat, performed the high-pressure experiments, and experimental analysis and participated in manuscript review and editing. Doreen C. Bayer synthetized ND3BD3 used for neutron experiments and participated in manuscript review and editing. Michael Hanfland as beamline scientist supported high-pressure experiments. Yuji Ikeda and Blazej Grabowski contributed to the modelling methodology and writing of the original draft and participated in manuscript review and editing. Ji Hun Yu, Sangsun Yang, Jai-Sung Lee, and Sergiy Divinski prepared and characterized starting alloy material and participated in manuscript review and editing. All authors discussed the results and approved the final manuscript.

## Funding

## Competing interests

The authors declare no competing interests.
