## [Transparent Peer Review file · Nature Communications]

Synthesis of High-Entropy Hydride from Cantor Alloy (fcc–CoCrFeNiMn) at Extreme Conditions

Corresponding Author: Dr Kirill Yusenko

Version 1:

Reviewer comments:

Reviewer #1

(Remarks to the Author)

The authors report the pressure-temperature conditions required to form a hydride from the Cantor alloy. They explore a range of pressure-temperature combinations and employ x-ray and neutron scattering for characterization, complemented by DFT calculations. The discovery of a new hydride phase and identification of the relevant synthesis window are valuable contributions to the fields of high-entropy systems and high-pressure physics. However, the conclusions regarding the Cantor alloy's resilience to hydrogenation are not strongly supported considering other publications that need further discussions. Below I provide detailed comments for revision:

1. Throughout the manuscript, "pressure" refers to the externally applied hydrostatic mechanical pressure. While this is an important metric for the phase transformation and reaction, in hydrogen materials research the hydrogen pressure or hydrogen chemical potential is equally critical, as it can guide the conditions of materials application to avoid unwanted reactions (such as hydrogen embrittlement) or promote desired reactions (such as hydride formation in hydrogen storage). The manuscript would benefit from a discussion of the hydrogen pressure/chemical potential actually accessed in these experiments and how this relates to hydride formation in the Cantor alloy.

2. The DAV01, DAC02, LVP01, and LVP02 experiments employ different compression, heating, and cooling settings. A clearer explanation of the rationale behind these variations is needed. What insights were the authors aiming to gain by comparing DAV01 vs DAV02; and LVP01 vs LVP02? What about DAV vs. LVP? From these comparisons, what can be concluded about the dominant factors driving hydride formation?

3. Similarly, when considering the impact of pressure and temperature to the formation of Cantor alloy hydride, the role of pressure is highlighted as demonstrated in the figures. However, the role of temperature is less clearly addressed.

a. Related to this, the statement that "compression with simultaneous heating forces HEH formation" appears counterintuitive, since hydride formation is generally promoted by higher pressure and lower temperature. A more careful discussion of how temperature influences hydrogen uptake in this system is needed.

4. The authors state that DFT calculations could not be extended to high pressures to probe hydrogen interstitial site preference. More explanation is needed: what specifically prevents such calculations in this system, given that several high-pressure hydride studies are available? Is this a technical limitation, or is it specific to the Cantor alloy system?

5. Presenting the pressure-temperature conditions required to form the Cantor alloy hydride, the authors concluded that the Cantor alloy is more resistant to hydrogenation than representative metals/alloys. To substantiate this claim, a more detailed comparison with representative systems should be provided. Additionally, it would also be valuable to discuss which specific factors (e.g., electronic structure, lattice strain, multicomponent effects) place the Cantor alloy in a different regime of hydrogenation resistance. This can benefit to design the conditions of potential application areas of the Cantor alloy.

6. Lastly, and most importantly, several prior studies have reported that the Cantor alloy ultimately undergoes mechanical degradation upon hydrogen insertion [Scripta Mater. 2017, 135, 54-58; Mater. Sci. Eng. A 2018, 736, 155-166; Mater. Sci. Eng. A 2018, 732, 105-111; Sci. Rep. 2017, 7(1), 9892]. Many of these experiments were performed by charging hydrogen and applying mechanical tension.

Furthermore, even when the hydrogen concentration (H/M) is significantly below 1 due to the use of relatively low hydrogen pressures, Bertsch et al. demonstrated that the Cantor alloy exhibits higher hydrogen solubility compared to conventional fcc alloys such as Ni and austenitic stainless steel 316L [Corros. Sci. 2021, 184, 109407]. More recently, Qian et al. reported insertion energies to make $H/M \approx 1$ is reasonably accessible [Comput. Mater. Sci. 2025, 252, 113789].

Taken together, these studies indicate that the Cantor alloy can, under accessible conditions, absorb significant amounts of hydrogen and suffer hydrogen-induced damage. The authors should reconcile their conclusion of exceptional hydrogenation resistance with this body of evidence, and explain the apparent discrepancy with prior experimental and computational findings when comparing the Cantor alloy with other representative systems.

Reviewer #2

(Remarks to the Author)

The manuscript by Glazyrin et al. carried out the synthesis of high-entropy hydride from the Cantor Alloy (fcc-CoCrFeNiMn) at extreme conditions. They report formation of an fcc hydride based on the Cantor alloy composition, and provide structural characterization, including an estimate of hydrogen content.

The current work focuses on two themes, hydrogen resistance and hydride synthesis, without a clearly defined central focus. The aspect of hydrogen resistance has already been partially addressed in the authors' study in 2024, while the hydride reported here cannot be stabilized at ambient pressure, which imposes significant limitations on its potential applications in hydrogen storage. The current work is not suitable for publication in Nature Communications in terms of scientific significance, practical applicability, and broad readership.

Reviewer #3

(Remarks to the Author)

This article addresses the synthesis of a high-entropy hydride derived from the Cantor alloy (fcc-CoCrFeNiMn) under extreme conditions. The finding that the Cantor alloy can indeed be hydrogenated in such conditions, with clear evidence of hydride formation, is particularly intriguing. Nevertheless, several aspects of the study could be strengthened, as the underlying mechanism has not been thoroughly explored. A more detailed discussion of this point would considerably enhance the manuscript.

Some comments that may be helpful to improve the manuscript are given below:

- a) Introduction: The motivation of this work should be clarified. Is there any particular application that necessitates pressures as high as 40 GPa? How does this study differ from the authors' previous work? What advantages does the Cantor alloy offer compared to conventional alloys such as 316L stainless steel in this context, since they may have similar mechanical properties and so on? I suggest that the author could think about the unique characteristics of HEAs.
- b) What is the specific novelty of synthesizing a hydride from the Cantor alloy? How does the difficulty of this synthesis compare to that of hydrides formed from conventional hydrogen-tolerant materials?
- c) I remain somewhat skeptical about the designation "high-entropy hydride." Can the authors be certain that a hydride phase derived from a high-entropy alloy should itself be classified as a high-entropy hydride?
- d) What role, if any, does configurational entropy play in the formation of the hydride phase? This point should be explicitly addressed.
- e) Figure 1: Why does heating result in a reduction in pressure? The statement "Comparison of the fcc-alloy signal in (a) with the fcc-hydride signal in (c) clearly indicates a larger unit cell volume for the latter" is not reader-friendly. How does the data support the claim of a larger unit cell volume? Are any calculations provided? Clearer guidance is needed, particularly for readers who are not XRD experts. The broadening of peaks attributed to the fcc-hydride phase should be interpreted more thoroughly. What physical insights can be drawn from this broadening? Figures 1d and 1e do not appear to provide meaningful information in their current form. If the authors wish to retain them, it may be more appropriate to place them in the Supplementary Information. For Fig. 1c, could the Au peaks be marked explicitly? Since Au is not central to this investigation, its contribution should be clarified.
- f) The resolution of Fig. 3b should be improved for better readability.
- g) What are the effects of hydrogen incorporation on the stability of the fcc structure?
- h) Numerous investigations have already explored the hydrogen embrittlement behavior of the Cantor alloy and related family alloys, where hydrogen charging has been confirmed (e.g., via TDS measurements). How does the hydride synthesized under extreme conditions in this work differ from the hydrogen-containing Cantor alloys previously studied?

i) The manuscript should articulate more clearly what unique advantages high-entropy alloys provide in this context compared to more conventional materials, as doing so could open new avenues for the exploration of the HEAs.

Version 2:

Reviewer comments:

Reviewer #1

(Remarks to the Author)

The authors properly addressed all questions.

Reviewer #2

(Remarks to the Author)

In this work, the authors have conducted very detailed theoretical and experimental analyses of the chemical interaction between the Cantor alloy and H₂, and have made extensions based on their previous studies. I have no doubts regarding the experimental results, however, compared to previous research, I think the authors have not made significant breakthroughs, particularly in terms of practical application significance. Therefore, I still consider that the current work is not suitable for publication.

Reviewer #3

(Remarks to the Author)

The manuscript, after the revisions made by the authors, has been thoroughly improved and is now suitable for acceptance.

REVIEWER COMMENTS

Reviewer #1 (Remarks to the Author):

The authors report the pressure-temperature conditions required to form a hydride from the Cantor alloy. They explore a range of pressure-temperature combinations and employ x-ray and neutron scattering for characterization, complemented by DFT calculations. The discovery of a new hydride phase and identification of the relevant synthesis window are valuable contributions to the fields of high-entropy systems and high-pressure physics.

We sincerely thank the Referee for the kind words and appreciation of our findings. We also realized that, as our work spans multiple areas of materials science and high-pressure research, some clarifications were necessary. We have revised the manuscript accordingly and believe the revisions have improved its readability and made it more accessible to the broad readership of Nature Communications.

However, the conclusions regarding the Cantor alloy's resilience to hydrogenation are not strongly supported considering other publications that need further discussions. Below I provide detailed comments for revision:

1. Throughout the manuscript, "pressure" refers to the externally applied hydrostatic mechanical pressure. While this is an important metric for the phase transformation and reaction, in hydrogen materials research the hydrogen pressure or hydrogen chemical potential is equally critical, as it can guide the conditions of materials application to avoid unwanted reactions (such as hydrogen embrittlement) or promote desired reactions (such as hydride formation in hydrogen storage). The manuscript would benefit from a discussion of the hydrogen pressure/chemical potential actually accessed in these experiments and how this relates to hydride formation in the Cantor alloy.

We thank the Referee for highlighting the importance of a clear differentiating the partial hydrogen pressure, i.e. the hydrogen chemical potential, from the total external pressure in evaluating hydride formation. In our DAC and LVP experiments, the externally applied pressure compresses hydrogen within a confined sample chamber, placing it in a regime of very high density. Under such conditions, the hydrogen fugacity, and therefore the hydrogen chemical potential, is strongly elevated (here we refer to P.L. Hagelstein "Equation of State and Fugacity Models for H₂ and for D₂." Journal of Condensed Matter Nuclear Science 2015 16 (1): 23–45. <https://doi.org/10.70923/001c.72356>) providing the thermodynamic driving force required for hydride formation once the appropriate pressure–temperature window is reached.

A straightforward experimental validation for $\mu(\text{H})$ in this environment is not available, because the chemical potential depends on several coupled factors, including the evolving hydrogen volume during reaction, the strongly non-ideal behaviour of dense hydrogen at multi-gigapascal pressures, and interactions with the sample and gasket, which prevent a reliable conversion from external pressure to $\mu(\text{H})$ in this geometry. Nevertheless, the extremely high hydrogen fugacity attained under the applied pressures provides the thermodynamic driving force consistent with the observed hydride formation.

In this framework, the Referee's point is well taken: the high applied pressure corresponds to a high and increasing hydrogen chemical potential, which enables hydride formation. We have revised the manuscript to clarify this point.

We introduced the following text addition on Page 5:

Although the hydrogen chemical potential cannot be easily extracted directly from DAC/LVP experiments, the strong confinement of hydrogen at multi-gigapascal pressures typically leads to increase of hydrogen fugacity (Ref: P.L. Hagelstein "Equation of State and Fugacity Models for H₂ and for D₂." Journal of Condensed Matter Nuclear Science 2015 16 (1): 23–45. <https://doi.org/10.70923/001c.72356>), providing the thermodynamic driving force for hydride formation.

2. The DAV01, DAC02, LVP01, and LVP02 experiments employ different compression, heating, and cooling settings. A clearer explanation of the rationale behind these variations is needed. What insights were the authors aiming to gain by comparing DAV01 vs DAV02; and LVP01 vs LVP02? What about DAV vs. LVP? From these comparisons, what can be concluded about the dominant factors driving hydride formation?

We thank the Referee for highlighting this point. We recognize that we did not explain the rationale behind the different experiments as clearly as they could have been in the original version, and we have now clarified it in the revision.

In the DAC experiments (DAV01 and DAV02), we varied the experimental trajectories in P-T space to probe thermodynamic boundaries. In addition, the similar outcomes of the two experiments also indicate that the formation process is primarily governed by the final pressure–temperature conditions rather than by the specific heating pathway.

In the LVP experiments (LVP01 and LVP02), the primary difference was the probing technique: LVP02 employed in-situ neutron diffraction to maximize hydrogen sensitivity, whereas LVP01 relied on X-rays. Two independent experiments yielded the same hydride phase, confirming that the observations are not dependent on the detection method and clarifying the position of the hydrogen atoms experimentally.

By combining DAC and LVP data, we leverage the complementary strengths of the two techniques: the DAC achieves higher pressures, while the LVP offers larger sample volumes and direct neutron sensitivity. The consistency across all four experiments shows that pressure–temperature conditions are the dominant factors controlling hydride formation, with differences in synthesis pathways, including heating rate, sample volume, or detection method, playing a secondary role.

We modified the manuscript in the following way:

Before we move to the conclusions, we would like to reiterate the following. In the DAC experiments (DAV01, DAV02), different P-T paths yielded the same results, indicating that hydride formation (including the final composition as reflected in the unit cell volume values) depends mainly on the final pressure–temperature conditions, not on the synthesis pathway. It is important to note that both temperature and pressure play important roles in the formation of the hydride. Indeed, both factors should affect hydrogen fugacity, and our compression at 100 °C shows that hydride forms at

moderate pressure. In contrast, at room temperature, the hydride does not form, even at the highest pressures applied in our study, whereas pure iron, for example, forms a hydride at ~3-4 GPa.

In addition, in the LVP experiments (LVP01, LVP02), the main difference was the probe: LVP02 used in-situ neutron diffraction, LVP01 used X-rays. Both produced the same hydride phase, confirming that the outcome is independent of the detection method and allowing experimental localization of hydrogen.

Combining DAC and LVP data contributes to redundancy and exploits their complementary strengths: higher pressures in DAC and larger volumes with neutron sensitivity in LVP. The agreement across all four experiments shows that final P-T conditions control hydride formation, while differences in synthesis pathways or techniques are secondary.

3. Similarly, when considering the impact of pressure and temperature to the formation of Cantor alloy hydride, the role of pressure is highlighted as demonstrated in the figures. However, the role of temperature is less clearly addressed.

We thank the Referee for raising this point. This comment encouraged us to re-examine the clarity with which the temperature dependence was presented in the original manuscript, and we have now expanded the corresponding explanations for greater clarity.

Our data indicate that temperature plays a significant role in promoting hydride formation. In the DAC experiments, for example, compression conducted at 100 °C shows that the hydride forms at substantially lower pressures compared with the maximum pressures of our current and previous studies at ambient temperature, where the hydride does not form at all. This behavior suggests that elevated temperature shifts the thermodynamic balance in favor of hydride formation. The additional experiment in which a pre-compressed DAC sample was subsequently heated in a vacuum oven provides independent confirmation of this effect.

Temperature also plays a critical role in the LVP experiments. Once the temperature is sufficiently high to release hydrogen from the precursor materials, the formation of the high-entropy hydride occurs readily, even at relatively low pressures. This trend is consistent across all LVP runs.

We have revised the text to make these points more explicit and to guide readers more clearly through the temperature-dependent behavior observed in both experimental scenarios. We hope that the expanded explanation resolves the Referee's concern and better highlights the combined influence of pressure and temperature on hydride formation.

a. Related to this, the statement that "compression with simultaneous heating forces HEH formation" appears counterintuitive, since hydride formation is generally promoted by higher pressure and lower temperature. A more careful discussion of how temperature influences hydrogen uptake in this system is needed.

We agree that in many conventional, thus, low-pressure experiments, hydrogenation could be promoted by the combination of high pressure and lower temperatures. In the high-pressure regime relevant to our DAC and LVP experiments, however, temperature plays an important kinetic and thermodynamic role. Elevated temperatures increase hydrogen mobility, accelerate diffusion through

the metal lattice, and can enable access to hydride stability fields that are not reached at ambient temperature. Numerous high-pressure studies report such behavior, including for metals such as Re and Fe. Note that high-pressure mitigate diffusion decreasing the activation energy Q via the $-P\Delta V$ term (with P being the external pressure and ΔV the activation volume of diffusion), but the increasing temperature accelerate diffusion significantly due to the exponential $\exp(Q/RT)$ dependence.

To clarify the concept, we provide an experimental example here: in DAC experiments using Re gaskets, hydride formation is minimal at room temperature, even at high pressure. However, moderate heating leads to rapid and readily detectable ReH_x formation (see the figure below). It is common for many metals that hydrogen dissociates at the metal surface and is subsequently absorbed into the bulk long-range lattice (material resistance plays some role, but this discussion is not necessary here). The absorbed hydrogen then diffuses through the metal. This behavior is consistent with the general trend in high-pressure hydrogenation, where elevated temperatures enhance hydrogen dissociation and mobility, enabling access to hydride stability fields that are not reached at ambient temperatures. Similar observations can be found in abundant literature. The same example with associated text and Figure is incorporated into the Supplementary:

The pure transition elements and their alloys can form hydrides. The formation of the associated hydrides is controlled by thermodynamic conditions and kinetics. Here we show a typical example of Re gasket loaded initially with a high entropy alloy and hydrogen in a DAC. The loading was compressed to 5 GPa at ambient temperature, it was then heated to 200 °C using a whole cell resistive heating setup of P02.2, PETRA-III, DESY and further compressed to 18 GPa. During this process, hydrogen dissociated at the boundary with Re and then diffused into the bulk of the metal. Similar is the process of hydrogenation of the Cantor alloy.

Microphotograph of a DAC loading with high-entropy hydride at 18 GPa and 200 °C showing Re gasket hydrogenation at elevated temperatures.

Significant diffusion of hydrogen through the gasket material leads to a collapse of the indicated sample chamber.

Formation of ReH_x within the gasket material can be easily verified using X-ray diffraction.

4. The authors state that DFT calculations could not be extended to high pressures to probe hydrogen interstitial site preference. More explanation is needed: what specifically prevents such calculations in this system, given that several high-pressure hydride studies are available? Is this a technical limitation, or is it specific to the Cantor alloy system?

We thank the Referee for pointing this out. We recognize that our previous text lacked clarity, which may have caused confusion. Indeed, the DFT calculations addressing partial occupancies at tetrahedral sites can be performed. We have clarified this point in the revised manuscript and have now carried out the corresponding calculations. The results involving partial tetrahedral occupation required additional computational time, and we have now completed supporting calculations. They are entirely consistent with the trends reported in the previous version of the manuscript.

Specifically, we compared a fully filled octahedral hydride with a half-filled (checkerboard-like) tetrahedral hydride, chosen to ensure the same overall hydrogen concentration in both configurations. The new results include: (a) the total energy per metal atom as a function of volume, (b) the enthalpy difference between octahedral and tetrahedral occupancies under pressure, and (c) the corresponding volume change as a function of pressure. A figure summarizing these calculations, together with explanatory text, has been added to the Supplementary Information and is also provided below.

As expected, the octahedral site occupation remains energetically favored, consistent with the dilute-limit results from Qian et al. (Comp. Mat. Sci., 2025). Furthermore, under applied pressure, octahedral sites remain the preferred choice. The equilibrium volumes for tetrahedral occupancy (at the same H concentration) are significantly larger, which is not observed experimentally, further confirming that the hydrogen protons prefer octahedral sites. We have revised the manuscript to include the additional DFT results, figure, and discussion on site preference under pressure in the Supplemental Material.

We added the following text to the manuscript:

We confirmed the octahedral site preference by performing additional DFT calculations for a half-filled (checkerboard) tetrahedral hydride, demonstrating that the octahedral arrangement remains energetically favored across all pressures (see Supplementary), consistent with recent dilute-limit calculations (Qian et al., 2025). Our computational results are also supported by the neutron scattering experiments described below.

The following text will be added to the Supplementary together with the supporting figure:

To clarify the hydrogen site preference at high pressures, additional DFT calculations were performed. Two distinct “half-filled” (checkerboard-like) tetrahedral hydride configurations (composition M:H, 1:1) were examined and compared with the octahedral configuration. Both tetrahedral configurations yielded essentially the same result, confirming the preference of dissociated hydrogen to octahedral sites. The results for one of them are summarized in Fig. S6, including: (a) total energy per metal atom versus volume per metal atom for octahedral and tetrahedral configurations, (b) the enthalpy difference between octahedral and tetrahedral occupancies under pressure, and (c) volume change as a function of pressure. The calculations indicate that the octahedral site remains energetically preferred at all pressures examined. The tetrahedral configurations exhibit substantially larger equilibrium volumes and higher total energies, which are inconsistent with experimental observations. This further confirms that the hydride adopts an octahedral occupancy.

5. Presenting the pressure-temperature conditions required to form the Cantor alloy hydride, the authors concluded that the Cantor alloy is more resistant to hydrogenation than representative metals/alloys. To substantiate this claim, a more detailed comparison with representative systems should be provided. Additionally, it would also be valuable to discuss which specific factors (e.g., electronic structure, lattice strain, multicomponent effects) place the Cantor alloy in a different regime of hydrogenation resistance. This can benefit to design the conditions of potential application areas of the Cantor alloy.

In principle, we appreciate the Referee's interest in obtaining a broader comparison across representative alloy systems and in identifying the underlying factors that govern hydrogenation resistance. However, providing a systematic, quantitative comparison with the wide range of relevant metals and alloys, as well as a comprehensive analysis of electronic structure, strain, and multicomponent effects, would require an extensive investigation that lies beyond the scope of the present study.

We note that several characteristics of the Cantor alloy, such as its significant chemical disorder, associated lattice distortions, and complex multicomponent electronic structure, have been identified in the literature as factors that can affect hydrogen uptake and diffusion. A comprehensive evaluation of these effects will require separate studies beyond the scope of the present work.

The diversity of materials, microstructures (including tempering state and grain size), chemical compositions (around the equiatomic one), and the pressure-temperature conditions relevant to hydrogenation spans a vast parameter space that cannot be comprehensively addressed within a single article. To offer context and to support further exploration, we refer the Referee to our recent work (doi: 10.1002/advs.202401741), cited in the manuscript, where hydrogenation behaviour in Cu-Be alloys and 316L steel is discussed in detail in the main text and Supplementary Information.

6. Lastly, and most importantly, several prior studies have reported that the Cantor alloy ultimately undergoes mechanical degradation upon hydrogen insertion [Scripta Mater. 2017, 135, 54-58; Mater. Sci. Eng. A 2018, 736, 155-166; Mater. Sci. Eng. A 2018, 732, 105-111; Sci. Rep. 2017, 7(1), 9892]. Many of these experiments were performed by charging hydrogen and applying mechanical tension. Furthermore, even when the hydrogen concentration (H/M) is significantly below 1 due to the use of relatively low hydrogen pressures, Bertsch et al. demonstrated that the Cantor alloy exhibits higher hydrogen solubility compared to conventional fcc alloys such as Ni and austenitic stainless steel 316L [Corros. Sci. 2021, 184, 109407]. More recently, Qian et al. reported insertion energies to make H/M

≈ 1 is reasonably accessible [Comput. Mater. Sci. 2025, 252, 113789].

Taken together, these studies indicate that the Cantor alloy can, under accessible conditions, absorb significant amounts of hydrogen and suffer hydrogen-induced damage. The authors should reconcile their conclusion of exceptional hydrogenation resistance with this body of evidence, and explain the apparent discrepancy with prior experimental and computational findings when comparing the Cantor alloy with other representative systems.

We thank the Referee for highlighting previous experimental studies on mechanical degradation, low-hydrogen absorption, and dilute-limit calculations. We appreciate these contributions and have acknowledged them in our revised manuscript. These studies indeed provide important context for understanding hydrogen–Cantor alloy interactions. At the same time, many of these works investigate different regimes, such as hydrogen charging under mechanical tension, higher-temperature autoclave conditions, or dilute-limit insertion energies, conditions important for preliminary material screening, but still far from actual material performance. Thus, they are therefore only partially comparable to the pressure–temperature conditions explored in the present study.

In our work, we introduce, for the first time, a Cantor-based hydride, representing the hydride counterpart of one of the most widely studied high-entropy alloys. This finding provides a novel and complementary perspective on earlier studies, offering new opportunities for investigating its mechanical and functional properties. While previous works suggested that hydrogen uptake in the Cantor alloy might occur under certain conditions, our results offer the first direct experimental confirmation of a stable hydride phase.

Furthermore, comparisons between dilute-limit calculations and concentrated hydride phases must be made cautiously, as the underlying thermodynamic and structural assumptions differ substantially. Nevertheless, where comparisons are possible, we observe no contradictions. For example, the dilute-limit octahedral site preference predicted by Qian et al. is fully consistent with both our DFT calculations and our experimental identification of an octahedral-based Cantor hydride.

Regarding the publication by Zhao et al. (Scripta Mater. 2017, 135, 54–58), we note that their conclusions include statements supportive of the Cantor alloy's relatively high resistance to gaseous hydrogen embrittlement compared with representative austenitic stainless steels. At the same time, we also observe that the composition in that study is reported only nominally, without accompanying microprobe characterization. In our work, the samples were carefully analyzed to verify their actual composition, which is crucial in multicomponent alloys, as even minor deviations can influence hydrogen–alloy interactions. In particular, in their Fig. 1 Zhao et al. show a SEM-BSE image with high recorded contrast. Because this contrast can be attributed to compositional variations, we simply note that our study includes additional microprobe measurements to verify composition, which is important for interpreting hydrogen uptake in multicomponent alloys. For the Referee's understanding only, we mention confidentially that preliminary, as-yet unpublished findings from our group indicate that deviations from the classical Cantor composition can reduce hydrogen resistance, further underscoring the importance of our results and the need for precise compositional control.

Concerning the work of Bertsch et al. (Corros. Sci. 2021), their experiments were conducted at moderately elevated temperatures (>200 °C), which significantly influence hydrogen solubility and mechanical response (see our response to point #3) above). These temperatures differ from the near-ambient regime central to our DAC and LVP measurements. Furthermore, their study focuses on a plasticity-mediated intergranular decohesion mechanism, while our work investigates bulk hydrogen

absorption and the thermodynamic stability of the hydride phase. These mechanisms are complementary but not directly comparable to each other. Therefore, we believe their conclusions cannot be straightforwardly transferred to our experimental conditions.

Lastly, our decompression experiments conducted using both DAC and LVP methods provide direct, experimentally grounded evidence that the hydride phase is thermodynamically unstable relative to the metallic Cantor alloy under the examined conditions. These measurements serve as a crucial experimental benchmark that complements and constrains theoretical models.

We have clarified these distinctions in the revised manuscript and hope that the additional explanations address the Referee's concerns.

Results & Discussion Chapter:

We confirmed the octahedral site preference by performing additional DFT calculations for a half-filled (checkerboard) tetrahedral hydride, demonstrating that the octahedral arrangement remains energetically favored across all pressures (see Supplementary), consistent with recent dilute-limit calculations (Qian et al., 2025). Our computational results are also supported by the neutron scattering experiments described below.

Conclusions:

The literature review including Refs (Refs Zhao et al 2017, Bertsch et al. 2021, Qian et al (2025)) highlights the complexity of studying the resistance of Cantor alloys to the multifaceted phenomenon of corrosion, including their permeability to hydrogen. This complexity may lead to interpretational inaccuracies, as individual studies often provide only a single perspective on a broader challenge. It should be emphasized that the answers in this field are far from straightforward. It is worth noting that some reports lack a thorough chemical analysis of the constituent elements, which is crucial for developing a comprehensive understanding. Moreover, results obtained from conventional methods (such as electrochemical and autoclave experiments) are not always directly comparable to each other or to the actual conditions under which practical components operate. At the same time, our results contribute an additional and important aspect to the overall multivector characterization by examining bulk hydrogen saturation in the material at specific temperatures, a perspective missing from conventional studies, which either typically employ higher temperatures to compensate for time limitations or from electrochemical approaches, which operate on an entirely different scale than methods described in our paper.

Reviewer #2 (Remarks to the Author):

The manuscript by Glazyrin et al. carried out the synthesis of high-entropy hydride from the Cantor Alloy (fcc-CoCrFeNiMn) at extreme conditions. They report formation of an fcc hydride based on the Cantor alloy composition, and provide structural characterization, including an estimate of hydrogen content.

The current work focuses on two themes, hydrogen resistance and hydride synthesis, without a clearly defined central focus. The aspect of hydrogen resistance has already been partially addressed in the authors' study in 2024, while the hydride reported here cannot be stabilized at ambient pressure, which imposes significant limitations on its potential applications in hydrogen storage. The

current work is not suitable for publication in Nature Communications in terms of scientific significance, practical applicability, and broad readership.

We thank the Referee for carefully reading our manuscript and for sharing their perspective. We respectfully disagree with the Referee's assessment of the scientific significance and scope of our work, and we appreciate the opportunity to clarify these points.

The present study provides two contributions that, in combination, have not appeared previously in the literature:

(1) the first experimental synthesis and structural characterization of a Cantor-based hydride, representing the hydride counterpart of one of the most widely studied high-entropy alloys, and

(2) a direct experimental assessment of hydrogen uptake and phase stability under controlled pressure–temperature conditions, which places the Cantor alloy's hydrogen response in a broader context relevant to both fundamental studies and practical environments.

These findings extend substantially beyond our earlier work and address aspects of hydrogen-Cantor alloy interactions that were previously inaccessible experimentally. While the hydride phase cannot be stabilized at ambient pressure, its formation pathway, site occupancy, thermal stability, and decompression behavior provide essential benchmarks for theory and open new directions for studying mechanical and functional properties under extreme conditions. Importantly, the relevance of such behavior is not limited to hydrogen-storage applications but also includes areas of the hydrogen economy involving structural materials, such as components exposed to high-pressure gaseous hydrogen, tubing, sealing elements, and reactors, where hydrogen stability, uptake, and release under load are critical design considerations.

To address the Referee's remark on focus and clarity, we have revised the manuscript to sharpen the narrative and explicitly articulate how hydride formation and hydrogen-resistance behaviour represent two linked aspects of the same underlying materials problem. Additional clarifications have been added to the Conclusions section to guide general readers through the complexity of hydrogen interactions in multicomponent alloys.

For completeness, we note that the included broader literature on Cantor alloys, such as Zhao et al. (2017), Bertsch et al. (2021), and Qian et al. (2025), highlights the multifaceted nature of hydrogen-related phenomena in these materials. Different studies naturally probe different regions of the hydrogen–Cantor alloy phase space, leading to varying observations. Our work explores a distinct, previously inaccessible high-pressure regime. Our work contributes a complementary and previously missing perspective by directly probing bulk hydrogen saturation and hydride thermodynamics, which cannot be accessed via electrochemical or autoclave methods. This multidimensional complexity is now emphasized more clearly in the revised manuscript.

We hope that the clarifications and revisions we have incorporated address the Referee's concerns and help convey the significance of our findings to the broad readership of Nature Communications.

Reviewer #3 (Remarks to the Author):

This article addresses the synthesis of a high-entropy hydride derived from the Cantor alloy (fcc–CoCrFeNiMn) under extreme conditions. The finding that the Cantor alloy can indeed be

hydrogenated in such conditions, with clear evidence of hydride formation, is particularly intriguing. Nevertheless, several aspects of the study could be strengthened, as the underlying mechanism has not been thoroughly explored. A more detailed discussion of this point would considerably enhance the manuscript.

We appreciate that the Referee finds the hydride formation in the Cantor alloy intriguing and agrees that the system is of broad interest. We also agree that a clearer description of the underlying mechanism can help guide the reader.

For clarity, we now briefly summarize the well-established sequence: molecular hydrogen dissociates at the metal surface, atomic hydrogen diffuses into the lattice under high pressure and elevated temperature, and progressive interstitial occupation leads to the expanded *fcc* unit cell observed in our diffraction data. Because Laue diffraction in both the DAC and LVP geometries probes the bulk of the compressed specimen, the measured increase in lattice parameters directly reflects bulk hydrogen incorporation.

To make this behavior more intuitive, we now include a widely recognized illustrative example: hydrogenation of Re gaskets in diamond-anvil-cell experiments. Under similar pressure–temperature conditions, ReH_x formation is readily detected by X-ray diffraction and can also be observed visually. We provide a micrograph of ReH_x formation at 18 GPa and 200 °C in the Supplementary Information, serving as a familiar reference for readers working with hydrogen under extreme conditions. The resubmitted Supplementary has the following text:

The pure transition elements and their alloys can form hydrides. The formation of the associated hydrides is controlled by thermodynamic conditions and kinetics. Here we show a typical example of Re gasket loaded initially with a high entropy alloy and hydrogen in a DAC. The loading was compressed to 5 GPa at ambient temperature, it was then heated to 200 °C using a whole cell resistive heating setup of P02.2, PETRA-III, DESY, and further compressed to 18 GPa. During this process, hydrogen dissociated at the boundary with Re and then diffused into the bulk of the metal. Similar is the process of hydrogenation of the bulk of the Cantor alloy.

Microphotograph of a DAC loading with high-entropy hydride at 18 GPa and 200 °C showing Re gasket hydrogenation at elevated temperatures.

Significant diffusion of hydrogen through the gasket material leads to a collapse of the indicated sample chamber.

Formation of ReH_x within the gasket material can be easily verified using X-ray diffraction.

We hope these additions and clarifications satisfy the Referee's request for a more precise explanation of the mechanism.

Some comments that may be helpful to improve the manuscript are given below:

a) Introduction: The motivation of this work should be clarified. Is there any particular application that necessitates pressures as high as 40 GPa? How does this study differ from the authors' previous work? What advantages does the Cantor alloy offer compared to conventional alloys such as 316L stainless steel in this context, since they may have similar mechanical properties and so on? I suggest that the author could think about the unique characteristics of HEAs.

We thank the Referee for these questions. In the revised manuscript, we clarify that the motivation of this work is twofold: (i) to synthesize and characterize, for the first time, the hydride derived from the Cantor alloy, and (ii) to map the pressure–temperature conditions that govern hydrogen incorporation. Pressures of up to 40 GPa are not directly motivated by applications. However, they are required to explore the phase stability fields, full site-occupancy landscape, and demonstrate that hydrogen remains confined to octahedral sites even at the highest pressures examined.

Compared to our previous work, the present study advances our understanding of the Cantor alloy hydrogen resistance into the range of elevated and industrially important temperatures. It also provides the first structural identification of the Cantor-based hydride, including its formation conditions and stability field.

Regarding the advantages of high-entropy alloys in general and Cantor alloy in particular relative to conventional alloys such as 316L, we note that HEAs offer a much higher portfolio of applications, due to their unique opportunities due to their chemical complexity, significant lattice distortions, and multifunctional character, which make them ideal platforms for exploring hydrogen incorporation and phase stability under extreme conditions. We clarified these points in the Introduction. In particular, we modified the text as follows:

This emerging field, at the intersection of chemistry, physics, and materials science, still holds significant potential for groundbreaking discoveries and applications. In comparison to conventional materials (e.g. 316L steel and others), the high entropy counterparts offer a broader range of applications not only because of their attractive mechanical, corrosion-resistant, and other properties, but also due to the large multicomponent phase space that provides greater freedom to tailor these properties. The relatively high costs of a bulk high-entropy CoCrFeMnNi alloy can motivate its application as hydrogen-facing coatings.

b) What is the specific novelty of synthesizing a hydride from the Cantor alloy? How does the difficulty of this synthesis compare to that of hydrides formed from conventional hydrogen-tolerant materials?

We thank the Referee for this important question. The novelty of synthesizing a hydride from the Cantor alloy lies in demonstrating that a prototypical high-entropy alloy (characterized by chemical complexity, severe lattice distortion, and multicomponent electronic structure) can form a well-defined hydride phase under high pressure. To our knowledge, this represents the first experimentally verified hydride derived from the Cantor alloy, placing it within the emerging but still underexplored class of high-entropy hydrides.

The difficulty of synthesizing such a phase reflects the intrinsic complexity of the alloy: hydrogen incorporation competes with significant local chemical disorder and lattice distortion, requiring higher pressures than in many conventional alloys. The resulting hydride provides new insight into how hydrogen interacts with chemically complex alloys and offers a valuable benchmark for theoretical models. It has always been challenging to quantify the difficulty for samples studied under ambient pressure conditions. They are typically correlated with residual content of hydrogen in specially prepared samples. Our results, including the previous publication (doi: 10.1002/ADVS.202401741 and related supplementary), can be used for comparison by examining the critical pressure at which a material (an alloy or a pure element) is transformed into a hydride at a given temperature.

c) I remain somewhat skeptical about the designation “high-entropy hydride.” Can the authors be certain that a hydride phase derived from a high-entropy alloy should itself be classified as a high-entropy hydride?

We thank the Referee for raising this point. High-entropy alloys, including the Cantor alloy, exhibit substantial configurational entropy due to the random distribution of multiple transition metals on a common sublattice. Although its precise stabilizing contribution in the Cantor alloy is still under active discussion, configurational entropy is broadly understood to stabilize a single-phase multicomponent solid solution at elevated temperatures which can be conditionally stable at application temperatures due to kinetic constraints.

Our DAC and LVP data show that, upon hydrogen incorporation, the metal sublattice remains chemically homogeneous and does not decompose into simpler binary or ternary hydrides. Consequently, the configurational entropy associated with the multicomponent metallic framework is preserved in the hydride phase. This adjusted entropy would be reduced if the alloy were to phase-separate, providing an indirect thermodynamic incentive for forming a single-phase Cantor-based hydride under the conditions studied.

At the same time, configurational entropy does not directly determine the occupancy of the hydrogen site. As our experimental and computational results show, hydrogen fully occupies the octahedral sites once the hydride forms, and this preference is governed by the local electronic and thermodynamic environment rather than by the entropy of the metal sublattice.

e) Figure 1: Why does heating result in a reduction in pressure? The statement “Comparison of the fcc-alloy signal in (a) with the fcc-hydride signal in (c) clearly indicates a larger unit cell volume for the latter” is not reader-friendly. How does the data support the claim of a larger unit cell volume? Are any calculations provided? Clearer guidance is needed, particularly for readers who are not XRD experts. The broadening of peaks attributed to the fcc-hydride phase should be interpreted more thoroughly. What physical insights can be drawn from this broadening? Figures 1d and 1e do not appear to provide meaningful information in their current form. If the authors wish to retain them, it may be more appropriate to place them in the Supplementary Information. For Fig. 1c, could the Au peaks be marked explicitly? Since Au is not central to this investigation, its contribution should be clarified.

We thank the Referee for these detailed suggestions. We have clarified several points in the revised manuscript to improve readability for a broader audience.

(i) Pressure decreases upon heating. (ii) Larger unit-cell volume of the hydride.

The reduction in pressure during heating is a standard effect in DAC experiments, reflecting the thermal relaxation of the whole diamond anvil cell as a whole mechanical assembly upon heating. We improved the manuscript by modifying the caption in the following way:

Fig.1 Selected powder X-ray diffraction patterns illustrating hydride formation. *The patterns were collected at the ID15B beamline, ESRF, with a wavelength of $\lambda = 0.4100 \text{ \AA}$ (Experiment DAC01) at ambient temperature. Phases including fcc-alloy, its hydride are indexed for clarity as well as contributions from the hcp-alloy and Au (pressure marker). The corresponding tick marks indicating different phases are shown under the patterns. Panel (a) shows the sample under compression in H_2 at ambient temperature. A DAC containing the sample was heated to $\sim 573 \text{ K}$ at an initial sample chamber pressure of $\sim 40.8(2) \text{ GPa}$, corresponding to panel (b). Then the DAC was cooled down to ambient temperature. Upon the heating-cooling cycle, the pressure in the sample chamber decreased, resulting from relaxation of the diamond anvil cell within the larger mechanical assembly. The resulting pattern is shown in (c). Comparison of the fcc-alloy signal in (a) with the fcc-hydride signal in (c) clearly indicates a larger unit cell volume for the latter. Here, a shift of diffraction peaks to lower 2θ directly corresponds to an increase in the lattice parameter via Bragg's law. Notably, the peaks attributed to the fcc-hydride phase exhibit broadening similar to the effect seen in pure metals. Panels (d) and (e) show 2D diffractograms corresponding to the panels (a) and (c), respectively, and show diffraction patterns collected using micro-beam.*

(iii) Calculations and data treatment.

The lattice parameters shown were obtained using standard least-squares refinement implemented in widely used crystallographic software packages (cited in the Methods section).

(iv) Peak broadening.

We believe that an extensive discussion of peak broadening would divert the focus away from the main topic, given the numerous techniques involved. We note that broadening due to hydrogenation is a typical feature even for pure metals. We also note that data collected with the micro-beam may not be suitable for a detailed analysis, as seen in Figures 1d and 1e. A comprehensive interpretation is beyond the scope of this work, and our preliminary analysis indicates that it would not provide significant additional insight to our major conclusions.

(v) Figures 1d and 1e.

We have chosen to retain these images because they provide raw diffraction information illustrating grain distribution and scattering characteristics of the compressed sample, which is valuable to readers, especially in the high-pressure community.

(vi) Au peaks.

Following the Referee's suggestion, we added information on the Au purpose as the pressure standard more clearly in the caption shown above.

We hope these revisions improve the clarity and accessibility of Fig. 1 for all readers.

f) The resolution of Fig. 3b should be improved for better readability.

We thank the Referee for the suggestion. In the final submission, Fig. 3b will be provided in high-resolution vector format to ensure optimal readability.

g) What are the effects of hydrogen incorporation on the stability of the fcc structure?

We thank the Referee for this question. In the revised manuscript, we now make this point more explicit. Our DFT calculations show that hydrogen incorporation modifies the relative stability of the fcc and hcp structures. Whereas increasing pressure would usually promote the hcp phase in the Cantor alloy, the formation of the hydride stabilizes the fcc-derived structure over the full range of pressures where the hydride is observed. This behavior is entirely consistent with our diffraction data, which show no evidence of an hcp hydride. Taken together, these computational and experimental results indicate that hydrogen insertion stabilizes the fcc-based hydride phase under the explored conditions.

h) Numerous investigations have already explored the hydrogen embrittlement behavior of the Cantor alloy and related family alloys, where hydrogen charging has been confirmed (e.g., via TDS measurements). How does the hydride synthesized under extreme conditions in this work differ from the hydrogen-containing Cantor alloys previously studied?

We appreciate the Referee for highlighting this important comparison. The hydrogen-containing Cantor alloys studied in previous autoclave or electrochemical experiments generally fall into the dilute hydrogen regime. In these studies, hydrogen is introduced at relatively low pressures, and the material is examined after it has been quenched. Under these conditions, hydrogen concentrations remain low, and the structural response is primarily influenced by trapping, defect interactions, and mechanical loading (experiment dependent), rather than by the formation of bulk phases.

In contrast, the present study accesses the high-concentration regime, where hydrogen fully occupies the octahedral interstitial sites and forms a distinct fcc hydride phase. This hydride is synthesized and probed in situ under controlled high pressure and temperature, allowing for the direct observation of its structure, stability, and decomposition pathway. This information cannot be obtained from conventional charging approaches.

Thus, the hydride reported here represents a different part of the hydrogen–Cantor alloy phase space: it is a thermodynamically stabilized, stoichiometric-like phase formed at high hydrogen chemical potentials, rather than a dilute solid solution. These two regimes are complementary but not directly comparable. For completeness, we also note that the hydride's decomposition upon unloading provides an experimental benchmark that is valuable for theoretical modeling of high-entropy hydrides.

To clarify this distinction for readers, we have included an expanded statement in the Conclusions section.

Change in Conclusion:

The literature review, including Refs (Zhao et al 2017, Bertsch et al. 2021, Qian et al (2025)), highlights the complexity of studying the resistance of Cantor alloys to the multifaceted phenomenon of corrosion, including their permeability to hydrogen. This complexity may lead to interpretational inaccuracies, as individual studies often provide only a single perspective on a broader challenge. It should be emphasized that the answers in this field are far from straightforward. It is worth noting

that some reports lack a thorough chemical analysis of the constituent elements, which is crucial for developing a comprehensive understanding. Moreover, results obtained from conventional methods (such as electrochemical and autoclave experiments) are not always directly comparable to each other or to the actual conditions under which practical components operate. At the same time, our results contribute an additional and important aspect to the overall multivector characterization by examining bulk hydrogen saturation in the material at specific temperatures, a perspective missing from conventional studies, which either typically employ higher temperatures to compensate for time limitations or from electrochemical approaches, which operate on an entirely different scale than methods described in our paper.

i) The manuscript should articulate more clearly what unique advantages high-entropy alloys provide in this context compared to more conventional materials, as doing so could open new avenues for the exploration of the HEAs.

We thank the Referee for this suggestion. In the revised Introduction and Conclusions, we now state more clearly that high-entropy alloys offer unique opportunities in this context due to their chemical complexity, lattice distortion, and multicomponent electronic structure, which distinguish them from conventional alloys such as 316L. These characteristics make HEAs particularly valuable for exploring hydrogen incorporation under extreme conditions. At the same time, we have maintained the manuscript's focus on the central aims of this study.

The introduction text was modified as follows:

This emerging field, at the intersection of chemistry, physics, and materials science, still holds significant potential for groundbreaking discoveries and applications. In comparison to conventional materials (e.g. 316L steel and others), the high entropy counterparts offer potentially a broader range of applications due to their exciting mechanical, corrosion resistance and multiple other properties.